# BRAINMOE: TOWARDS UNIVERSAL EEG FOUNDATION MODELS WITH CHANNEL-WISE MIXTURE-OF-EXPERTS

## ABSTRACT

Electroencephalography (EEG) is pivotal for brain-computer interface (BCI) and healthcare applications, with foundation models offering a promising paradigm for generalized decoding. However, current models typically apply a uniform strategy across all channels, overlooking the brain's inherent functional heterogeneity. This "*one-size-fits-all*" approach limits model adaptability and performance, as different EEG tasks activate distinct neural pathways. To address this, we introduce BRAINMOE, a novel universal EEG foundation model featuring a *channel-wise Mixture-of-Experts (MoE)* architecture. The core innovation is the dynamic assignment of specialized experts to different channels for more nuanced decoding. BRAINMOE first employs a *ChannelFormer* to capture distinct representations for each channel. A router then uses these learned representations to intelligently select and weight the most relevant experts. This mechanism allows BRAIN-MOE to tailor its decoding strategy on a per-channel basis, aligning its computation with the varied spatial demands of diverse EEG tasks. Comprehensive experiments on seven downstream tasks across nine benchmarks demonstrate that BRAINMOE achieves significant performance gains, setting a new state-of-the-art on six datasets. These results validate that channel-wise specialization is a critical step towards more powerful and truly universal EEG foundation models, showcasing the robust capability and generalizability of our approach.

## 1 INTRODUCTION

Brain-Computer Interfaces (BCI) are systems that establish a direct communication pathway between the human brain and external devices, enabling novel forms of interaction and control (Schalk et al., 2004; Zhang et al., 2019). Electroencephalography (EEG) is crucial for BCI due to its ability to provide a real-time, portable, and accessible window into brain function, which can be leveraged to interpret a user's intentions or cognitive states. In recent years, researchers have developed algorithms to decode a wide array of brain activities, including emotion recognition (Dadebayev et al., 2022; Gao et al., 2024), motor imagery classification (Dai et al., 2020; Altaheri et al., 2021), epileptic seizure detection (Ahmad et al., 2022; Yıldız et al., 2022), and sleep staging (Wang et al., 2024b; Zhou et al., 2024).

Inspired by the success of self-supervised learning and foundation models in computer vision (Dosovitskiy et al., 2020) and natural language processing (Devlin et al., 2018; OpenAI, 2023), the neuroscience community has begun developing EEG foundation models (Kostas et al., 2021; Chien et al., 2022; Yang et al., 2023; Jiang et al., 2024; Wang et al., 2024a). These models are pre-trained on large quantities of unlabeled EEG data to learn robust and generalizable representations, which can then be fine-tuned for various BCI applications such as motor imagery classification, seizure detection, and cognitive workload monitoring (Wang et al., 2025). Most existing EEG foundation models treat the input signal as a uniform data structure, seg-

menting it into patches across channels and time. A monolithic architecture, *e.g.*, a standard Transformer, is then used to model the dependencies among all patches simultaneously.

However, this "one-size-fits-all" approach overlooks a critical characteristic of brain activity: **functional heterogeneity**. Different EEG tasks activate distinct neural circuits (Razoumnikova, 2000; Tavor et al., 2016), meaning the importance and informational content of each EEG channel can vary significantly depending on the specific cognitive or motor task being performed. For instance, motor imagery tasks often rely on channels over the motor cortex, whereas working memory tasks emphasize frontal and parietal channels (Figueroa-Vargas et al., 2020; Chen et al., 2021). Current foundation models apply the same decoding strategy and computational resources to every channel, regardless of its relevance to the task at hand. This uniform processing is suboptimal and fails to account for the specialized functions of different brain regions, potentially limiting the model's adaptability and overall performance.

To address this fundamental limitation, we introduce BRAINMOE, a novel universal EEG foundation model that leverages a *channel-wise Mixture-of-Experts (MoE)* architecture. Instead of using a single, monolithic model for all channels, BRAINMOE dynamically assigns specialized processing experts, to decode information from different channels. Our approach first employs a *ChannelFormer* to generate the representation for each EEG channel. Specifically, we adopt learned query tokens to attend all patches in a channel via the cross-attention to produce the channel representation. For EEG signals of arbitrary length, the ChannelFormer can always output a fixed-size channel embedding by learning patterns from all patches. This representation is then passed to a router, which intelligently selects the top-$K$ most appropriate experts for that specific channel and calculates their contribution weights. This mechanism allows BRAINMOE to allocate specialized computational pathways based on the unique information content of each channel, creating a highly adaptive and efficient decoding process.

To summarize, our main contributions are three-fold:

- We introduce BRAINMOE, a novel EEG foundation model that employs a *channel-wise Mixture-of-Experts (MoE)* architecture. By dynamically routing channels to specialized experts, our model adapts its encoding strategy to the unique functional demands of diverse BCI tasks.

- We propose a novel *ChannelFormer* module that learns distinct channel-level representations to guide the expert routing mechanism. This enables an intelligent, adaptive allocation of computational resources, moving beyond the rigid, one-size-fits-all paradigm of previous models.

- We conduct comprehensive evaluations on nine diverse EEG benchmarks, demonstrating that BRAINMOE significantly outperforms existing methods and achieves state-of-the-art performance on six datasets. These results validate the effectiveness of channel-wise specialization and establish our model's robust generalization capabilities for universal EEG encoding.

## 2 METHOD

**Problem Statement.** Given an EEG sample denoted as $\mathbf{S} \in \mathbb{R}^{C \times T}$, where $C$ is the number of electrode channels and $T$ is the number of timestamps, our goal is to learn a universal representation model. This model is pre-trained on a large corpus of unlabeled EEG data to capture robust spatial-temporal dependencies. To handle the diverse formats of EEG signals (*e.g.*, varying channels and lengths) in real-world applications, we first segment the raw sample $\mathbf{S}$ into a set of fixed-size patches $\mathbf{X} \in \mathbb{R}^{C \times n \times t}$, where $n = \lfloor \frac{T}{t} \rfloor$ is the number of patches per channel and $t$ is the patch length. The objective of pre-training is to optimize the model parameters such that the learned representations are generalizable and can be effectively fine-tuned for a wide range of downstream BCI tasks.

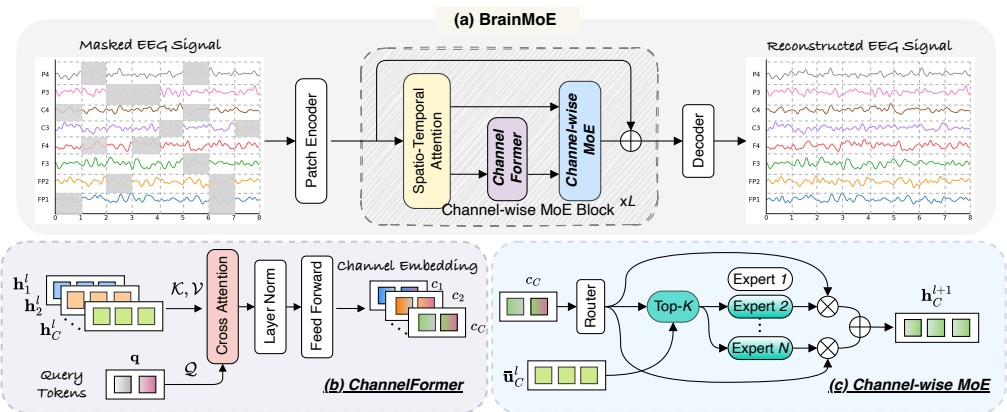

Figure 1: The architecture of BRAINMOE with (b) *ChannelFormer* and (c) *Channel-wise MoE* mechanism.

## 2.1 BRAINMOE ARCHITECTURE OVERVIEW

The core idea of BRAINMOE is to process EEG signals with a channel-wise MoE architecture, which dynamically allocates specialized computational resources based on the functional characteristics of each channel. As depicted in Figure 1, the overall pre-training architecture comprises three main stages: 1) a patch encoder that converts the raw EEG signal into a sequence of initial feature representations, 2) a stack of our novel channel-wise MoE blocks that process these features to learn powerful latent representations, and 3) a decoder that reconstructs the masked portions of the original signal from these final representations.

**Patch Encoder.** Following standard practice in recent EEG foundation models (Song et al., 2022; Jiang et al., 2024), we first segment the raw EEG signal into a sequence of fixed-size patches, $\mathbf{X} = \{\mathbf{x}_{i,j} | i \in [1, \ldots, C], j \in [1, \ldots, n]\}$, where $\mathbf{x}_{i,j} \in \mathbb{R}^t$. For our self-supervised pre-training objective, a random subset of these patches is masked. The sequence of patches is then passed through a patch encoder which extracts local features by incorporating both temporal and frequency-domain information, consistent with prior work (Wang et al., 2025). The encoder outputs a sequence of patch embeddings, $\{\mathbf{e}_{i,j}\} \in \mathbb{R}^d$, which subsequently serve as the input to the channel-wise MoE blocks.

**Channel-wise MoE Block.** We replaces the standard FFN sub-layer with a channel-wise MoE layer. For a given channel $i$, all its constituent patch embeddings are processed by the same set of selected experts. The router first take the channel representation as input, calculates softmax gating scores over the $N$ available experts. We employ Top-$K$ routing, where only the experts with the $K$ highest scores are activated for that channel. The operation of the Channel-wise MoE block for any patch embedding $\mathbf{h}_{i,j}^l$ from channel $i$ within layer $l$ is defined as:

$$\mathbf{u}_{i,j}^l = \text{STA}\left(\text{LayerNorm}\left(\mathbf{h}_{i,j}^l\right)\right) + \mathbf{h}_{i,j}^l, \tag{1}$$

$$\bar{\mathbf{u}}_{i,j}^l = \text{LayerNorm}\left(\mathbf{u}_{i,j}^l\right), \tag{2}$$

$$\mathbf{h}_{i,j}^{l+1} = \text{ChannelMoE}\left(\bar{\mathbf{u}}_{i,j}^l, \mathbf{c}_i\right) + \mathbf{u}_{i,j}^l, \tag{3}$$

where STA represents the spatial-temporal multi-head attention mechanism in CBraMod (Wang et al., 2025), $\mathbf{h}_{i,j}^l$ is the patch feature output by last block and $\mathbf{h}_{i,j}^0 = \mathbf{e}_{i,j}$. The $\mathbf{c}_i \in \mathbb{R}^d$ denotes the embedding for the $i^{th}$ channel (see details in **ChannelFormer** below). The ChannelMoE$(\cdot)$ is defined as a weighted sum of

expert outputs:

$$\text{ChannelMoE}(\mathbf{x}, \mathbf{c}_i) = \sum_{k=1}^{N} g(\mathbf{c}_i)_k \cdot \text{Expert}_k(\mathbf{x}) + \text{Expert}_{N+1}(\mathbf{x}), \tag{4}$$

$$g(\mathbf{c}_i)_k = \begin{cases} \frac{e^{\mathbf{W}_k \mathbf{c}_i}}{\sum_{j \in \mathcal{T}} e^{\mathbf{W}_j \mathbf{c}_i}}, & k \in \mathcal{T}, \\ 0, & \text{otherwise.} \end{cases} \tag{5}$$

where $\mathbf{W}_j$ is the learnable weight matrix of the router, $\mathcal{T}$ denotes the set of indices for the top-$K$ scores, and each of the $N$ specialized experts, $\text{Expert}_k(\cdot)$, is a standard feed-forward network. A shared expert, $\text{Expert}_{N+1}$, is also activated for all channels to capture common features. This channel-wise routing ensures that specialized experts are engaged for different channels, thereby aligning the model's computation with the functional heterogeneity of the brain.

**ChannelFormer.** A key innovation of BRAINMOE is its adaptive processing of each channel based on its functional role. To enable this, we introduce the ChannelFormer, a module designed to generate a comprehensive representation for each channel. While a simple approach like mean-pooling could aggregate temporal information, it risks losing critical patterns within the signal.

Therefore, as illustrated in Figure 1, our ChannelFormer adopts a more sophisticated attention-based mechanism. We employ a set of $m$ learnable query tokens, $\mathbf{q} \in \mathbb{R}^{m \times d}$. For the $i$-th channel, these tokens attend to the full sequence of its patch embeddings, $\mathbf{h}_i = \{\mathbf{h}_{i,0}, \ldots, \mathbf{h}_{i,n}\}$, which serve as the key and value. The output of this cross-attention operation is then processed through a layer normalization and an FFN to produce the final channel embedding:

$$\mathbf{c}_i = \text{FFN}(\text{LayerNorm}(\text{MSA}(\mathbf{q}, \mathbf{h}_i, \mathbf{h}_i))), \tag{6}$$

where $\text{MSA}(\cdot)$ denotes multi-head self-attention. The resulting set of channel embeddings, $\{\mathbf{c}_i\}_{i=1}^{C}$, provides a rich, condensed summary of each channel's activity, which is subsequently used by the router in each MoE layer to guide expert selection. A core advantage of this design is its ability to handle EEG signals of arbitrary length, as the attention mechanism can distill information from any number of input patches into a fixed-size channel embedding.

## 2.2 PRE-TRAINING OBJECTIVES

The pre-training of BRAINMOE is guided by a combination of a primary reconstruction objective and an auxiliary loss to ensure balanced expert utilization.

**Masked EEG Reconstruction.** We adopt the Masked Autoencoder (MAE) framework (He et al., 2022) for self-supervised pre-training. A significant portion of the input EEG patches are randomly masked with zeros. The model is then tasked with reconstructing the original masked patches from the latent representations of the visible unmasked patches. A lightweight linear decoder is added on top of the final Transformer block to predict the patch values. The reconstruction loss is calculated as the Mean Squared Error (MSE) between the reconstructed and original patches, but only over the masked set:

$$\mathcal{L}_{\text{recon}} = \frac{1}{|\mathcal{M}|} \sum_{(i,j) \in \mathcal{M}} \|\mathbf{x}_{i,j} - \hat{\mathbf{x}}_{i,j}\|_2^2, \tag{7}$$

where $\mathcal{M}$ is the set of indices of the masked patches, $x_{i,j}$ is the original patch, and $\hat{x}_{i,j}$ is the reconstructed patch.

**Expert Balance Loss.** A common challenge in training MoE models is "routing collapse" where the router consistently favors a small subset of experts. To encourage a balanced load across all experts, we incorporate an auxiliary load-balancing loss, adapted from recent MoE training work (Dai et al., 2024). This loss incentivizes the router to distribute channels evenly among the experts. The auxiliary loss is defined as:

$$\mathcal{L}_{\text{aux}} = N \sum_{k=1}^{N} f_k \cdot p_k, \tag{8}$$

where $f_k$ is the fraction of channels routed to expert $k$ within a batch, and $p_k$ is the average router probability assigned to expert $k$ across all channels in the batch.

**Total Loss.** The final pre-training objective for BRAINMOE is a weighted sum of the reconstruction loss and the auxiliary balance loss:

$$\mathcal{L} = \mathcal{L}_{\text{recon}} + \alpha \mathcal{L}_{\text{aux}}, \tag{9}$$

where $\alpha$ is a hyperparameter for the balance loss. The combined objective trains BRAINMOE to learn powerful EEG representations while ensuring all specialized experts are effectively utilized.

## 3 EXPERIMENTS

### 3.1 EXPERIMENTAL SETUP

**Pre-training Data and Implementation Details.** To ensure robust and generalizable representation learning, we pre-train our BRAINMOE on the Temple University Hospital (TUEG) EEG corpus (Obeid & Picone, 2016), the largest publicly available clinical EEG dataset. Following preprocessing protocols (Jiang et al., 2024; Wang et al., 2024a), the EEG signals are band-pass filtered between 0.5-75 Hz and notch-filtered at 60 Hz to remove power-line noise, then resampled to 200 Hz and segmented into non-overlapping 30-second windows. After cleaning, the corpus yields over one million EEG segments for pre-training. Pre-training is conducted for 40 epochs with batch size 512 on 4 NVIDIA H200 GPUs. We set $\alpha = 0.01$ for the expert balance loss and 50% masking ratio for EEG reconstruction. Additional training details are provided in the Appendix.

**Downstream Tasks and Datasets.** We assess the generalization ability of BRAINMOE across a diverse set of BCI tasks, including *Sleep Staging* (*e.g.*, ISRUC (Khalighi et al., 2016)), *Emotion Recognition* (*e.g.*, SEED-V (Liu et al., 2021), FACED (Zhang et al., 2024)), *Motor Imagery* (*e.g.*, PhysioNet-MI (Schalk et al., 2004), SHU-MI (Ma et al., 2022)), *Seizure Detection* (*e.g.*, CHB-MIT (Shoeb, 2009)), *Imagined Speech* (*e.g.*, BCIC2020-3 (Jeong et al., 2022)), *Event Classification* (*e.g.*, TUEV (Obeid & Picone, 2016)), and *Mental Stress Detection* (*e.g.*, MentalArithmetic (Goldberger et al., 2000)).

**Baselines and Evaluation Metrics.** We compare BRAINMOE against serveal state-of-the-art methods, grouped into two categories: *1) Task-specific models*, including EEGNet (Lawhern et al., 2018) and Conformer (Song et al., 2022); and *2) EEG foundation models*, such as BIOT (Yang et al., 2023), LaBraM (Jiang et al., 2024), and CBraMod (Wang et al., 2025). For evaluation, we report Balanced Accuracy (B-Acc) and Area Under the Receiver Operating Characteristic (AUROC) on binary tasks, and B-Acc with weighted F1 (F1-W) on multi-class tasks. All results are averaged over five runs with different random seeds.

### 3.2 COMPARISON WITH STATE-OF-THE-ART METHODS

Table 1 presents a comprehensive performance comparison of our proposed BRAINMOE against a suite of representative task-specific and foundation models across nine diverse BCI tasks. The results clearly highlight the superiority of our approach, leading to two primary conclusions:

Table 1: **Performance comparison** of various models across nine EEG datasets. We **bold** the best performance and underline the second-best performance.

| Dataset | Metric | EEGNet | Comformer | CNN-Trans | FFCL | ST-Trans | BIOT | LaBraM | CBraMod | CodeBrain | Ours |
|---------|--------|--------|-----------|-----------|------|----------|------|--------|---------|-----------|------|
| FACED | B-Acc | 40.90 | 45.59 | 46.97 | 46.73 | 48.10 | 51.18 | 52.73 | 55.09 | 58.16 | **63.13** |
| | F1-W | 41.24 | 45.14 | 47.02 | 46.99 | 47.95 | 51.36 | 52.88 | 56.18 | 58.49 | **63.34** |
| SEED-V | B-Acc | 29.61 | 35.37 | 36.78 | 36.41 | 30.52 | 38.37 | 39.76 | 40.91 | 41.37 | **41.90** |
| | F1-W | 27.49 | 34.87 | 36.42 | 36.45 | 28.33 | 38.56 | 39.74 | 41.01 | 42.35 | **42.82** |
| SHU-MI | B-Acc | 58.89 | 59.00 | 59.75 | 56.92 | 59.92 | 61.79 | 61.66 | 63.70 | **64.31** | 63.97 |
| | AUROC | 62.83 | 63.51 | 63.43 | 63.26 | 64.31 | 66.09 | 66.04 | 69.88 | **71.24** | 70.34 |
| TUEV | B-Acc | 38.76 | 40.74 | 40.87 | 39.79 | 39.84 | 52.81 | 64.09 | 66.71 | 64.28 | **68.53** |
| | F1-W | 65.39 | 69.83 | 68.54 | 67.83 | 68.23 | 74.92 | 83.12 | 83.42 | 83.62 | **85.23** |
| ISRUC | B-Acc | 71.54 | 74.00 | 73.63 | 72.77 | 73.18 | 75.27 | 76.33 | **78.65** | 78.35 | 78.35 |
| | F1-W | 75.13 | 76.34 | 77.19 | 76.14 | 76.81 | 77.90 | 78.10 | **80.11** | 80.02 | **80.11** |
| CHB-MIT | B-Acc | 56.58 | 59.76 | 63.89 | 62.62 | 59.15 | 70.68 | 70.75 | **73.98** | 72.73 | 73.62 |
| | AUROC | 80.48 | 82.26 | 86.62 | 82.71 | 82.37 | 87.61 | 86.79 | 88.92 | 89.61 | **89.99** |
| BCIC-2020 | B-Acc | 44.13 | 45.06 | 45.33 | 46.78 | 41.26 | 49.20 | 50.60 | 53.73 | 61.01 | **64.01** |
| | F1-W | 44.13 | 44.88 | 45.06 | 46.89 | 42.47 | 49.17 | 50.54 | 53.83 | 61.01 | **64.00** |
| PhysioNet-MI | B-Acc | 58.14 | 60.49 | 60.53 | 57.26 | 60.35 | 61.53 | 61.73 | 61.74 | 61.87 | **65.28** |
| | F1-W | 57.96 | 60.62 | 60.41 | 57.01 | 60.53 | 61.58 | 61.77 | 61.79 | 61.86 | **65.53** |
| MentalArithmetic | B-Acc | 67.70 | 68.05 | 67.79 | 67.98 | 66.31 | 68.75 | 69.09 | 72.56 | 75.14 | **75.34** |
| | AUROC | 73.21 | 74.24 | 72.58 | 73.30 | 71.32 | 75.36 | 77.21 | 79.05 | **87.07** | 86.16 |
| **Avarage** | | 51.81 | 54.23 | 55.06 | 54.14 | 53.18 | 58.84 | 60.75 | 63.01 | 64.14 | **66.01** |

**1) BRAINMOE establishes a new state-of-the-art across a majority of the evaluated BCI tasks.** The model demonstrates particularly significant performance gains on complex decoding benchmarks. For instance, on the imagined speech task (BCIC-2020), BRAINMOE surpasses the leading foundation models, CBraMod and CodeBrain, by substantial margins in B-Acc (+10.28% and +3.00%, respectively). Similar large improvements are observed on the emotion recognition task (FACED), with B-Acc gains of +8.04% and +4.97%. These results underscore the robust capability of our approach to decode subtle and complex brain states more effectively than existing methods.

**2) The channel-wise MoE architecture provides a substantial performance advantage.** The performance gains can be attributed to our novel architecture. By dynamically assigning specialized experts to different EEG channels, BRAINMOE effectively captures the functional heterogeneity of neural signals—a capability that models with uniform channel processing lack. This advantage is evident in its performance on TUEV, where it achieves a +4.44% B-Acc improvement over LaBraM, and on PhysioNet-MI, with a +3.54% gain over CBraMod. These consistent improvements validate our central hypothesis that modeling channel-specific dynamics is a critical design principle for building the next generation of robust and universal EEG foundation models.

### 3.3 ABLATION STUDY

To gain more insights into BRAINMOE, we conduct a set of ablative studies on four BCI tasks.

**Effect of channel-wise MoE.** We first investigate the essential designs in BRAINMOE, *i.e.*, channel-wise MoE. We evaluate the necessity of our core architectural designs by comparing the full BRAINMOE model against two variants: 1) a Non-MoE baseline where the MoE layer is replaced with a standard FFN, and 2) a

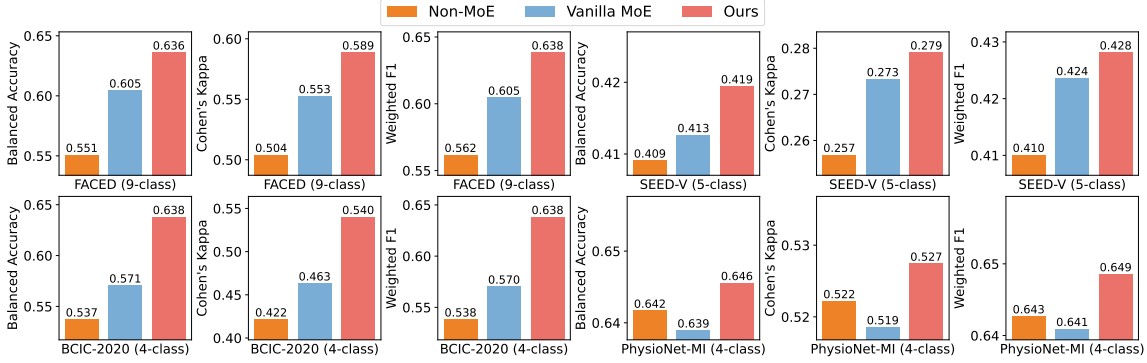

Figure 2: **Performance comparison of MoE variants** across four EEG datasets.

Table 2: **Performance comparison with different query numbers** across four EEG datasets.

| Dataset | Mean | | | 1 Query | | | 2 Query | | | 4 Query | | |
|---|---|---|---|---|---|---|---|---|---|---|---|---|
| | B-Acc | Kappa | F1-W | B-Acc | Kappa | F1-W | B-Acc | Kappa | F1-W | B-Acc | Kappa | F1-W |
| FACED | 62.94 | 58.10 | 62.98 | 61.80 | 56.79 | 61.89 | 61.98 | 56.92 | 63.01 | 63.62 | 58.88 | 63.85 |
| SEED-V | 41.41 | 27.81 | 42.27 | 41.78 | 27.69 | 42.76 | 41.45 | 27.31 | 42.31 | 41.95 | 27.91 | 42.82 |
| BCIC-2020 | 62.00 | 52.50 | 61.95 | 64.20 | 55.10 | 64.08 | 63.07 | 53.83 | 63.08 | 63.80 | 54.02 | 63.83 |
| PhysioNet-MI | 64.23 | 52.30 | 64.37 | 63.78 | 51.71 | 64.05 | 64.34 | 52.45 | 64.48 | 64.56 | 52.74 | 64.86 |
| **Average** | 57.64 | 47.68 | 57.68 | 57.89 | 47.82 | 58.20 | 57.71 | 47.63 | 57.97 | 58.48 | 48.39 | 58.84 |

vanilla MoE version, where each patch selects its own experts independently via the router. All experiments are conducted on four datasets: FACED, SEED-V, BCIC-2020, and PhysioNet-MI. As shown in Figure 2, our proposed channel-wise MoE architecture significantly outperforms both baseline variants across all datasets.

On average, our model achieves a B-Acc of $58.5\%$, surpassing the Non-MoE baseline by $+5.0\%$ and the vanilla MoE by $+2.8\%$. This highlights two key findings: *First*, simply replacing the FFN with a standard MoE layer (*i.e.*, vanilla MoE) yields noticeable but limited gains, confirming the general benefit of expert specialization. *Second*, and more critically, enforcing channel-wise routing provides a substantial additional performance boost. The most dramatic improvement is seen on the BCIC-2020 dataset, where channel-wise MoE improves B-Acc by nearly $7\%$ over vanilla MoE ($63.8\%$ vs. $57.1\%$). This validates our hypothesis that treating all temporal patches from a single channel as a coherent unit for expert routing is essential for effectively modeling the distinct functional roles of different brain regions. The results confirm that both the MoE structure and our channel-aware routing mechanism are vital to the success of BRAINMOE.

**Design of ChannelFormer.** We next examine the design of our ChannelFormer. The purpose of ChannelFormer is to produce the channel embedding, which is then used to select experts for each channel. Here, the baseline approach uses mean pooling over all patches within the same channel to generate the channel representation. We compare this against our ChannelFormer design using a varying number of learnable query tokens, specifically setting the number of tokens to 1, 2, and 4.

As shown in Table 2, all variants utilizing learnable query tokens outperform the simple mean pooling baseline on average, demonstrating the effectiveness of learning an adaptive representation for each channel. The configuration with 4 query tokens achieves the best overall performance, with an average B-Acc of

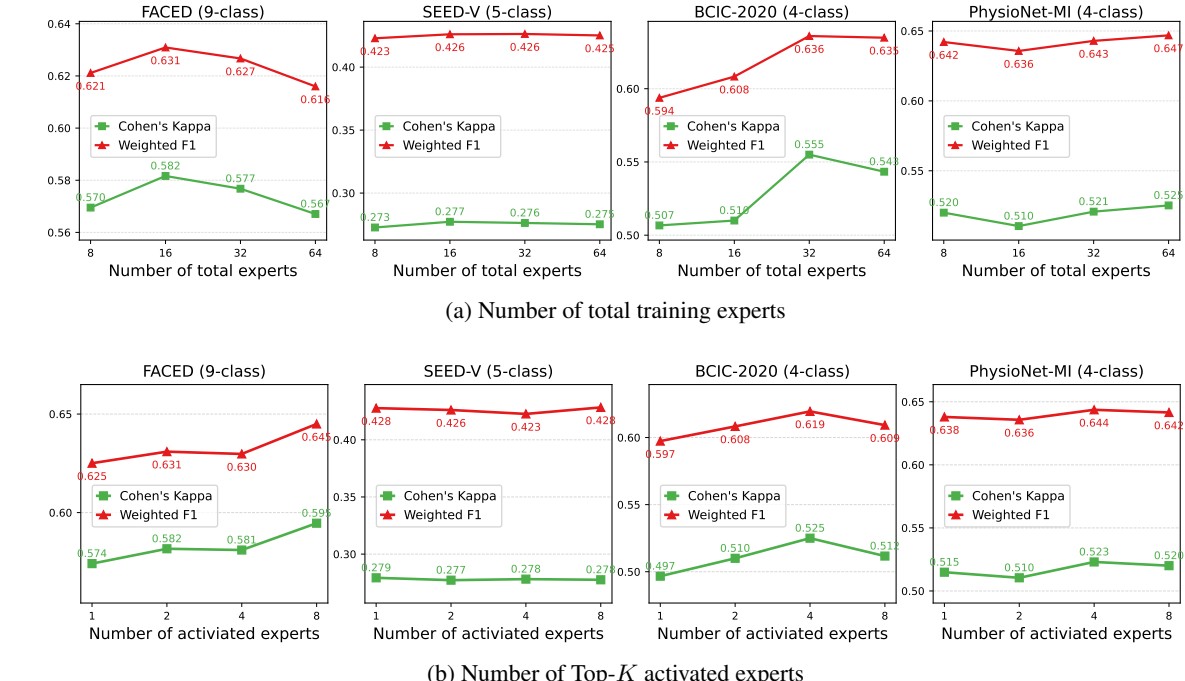

(a) Number of total training experts

(b) Number of Top-$K$ activated experts

Figure 3: **Study of the Mixture-of-Experts (MoE) configuration.** Performance across four datasets is shown while varying (a) the total number of experts ($N$) with fixed $K = 2$, and (b) the number of top-$K$ activated experts with fixed $N = 16$.

58.48%, surpassing the mean pooling baseline by nearly 1%. This suggests that employing multiple query tokens allows the model to capture a richer and more comprehensive representation of channel-specific characteristics, leading to more effective expert routing. Interestingly, even using a single learnable query token yields a noticeable improvement over mean pooling, particularly on the BCIC-2020 dataset where it achieves the highest B-Acc. This highlights that a learned attention mechanism is superior to a static aggregation method. Given its robust and superior performance across multiple datasets, we adopt the 4-query configuration for the ChannelFormer in our final BRAINMOE architecture.

**Model Scaling with Experts.** We analyze the impact of two key hyperparameters in our channel-wise MoE architecture: the total number of experts available during training ($N$) and the number of experts activated for routing at inference time ($K$). This analysis, illustrated in Figure 3, sheds light on the trade-offs between model capacity, computational cost, and performance.

Figure 3a shows the performance as we scale the total number of experts ($N \in \{8, 16, 32, 64\}$) while keeping the number of activated experts fixed at $K = 2$. The results indicate that increasing the expert pool generally improves or stabilizes performance, though the effect varies by dataset. The most pronounced benefit is observed on the BCIC-2020 dataset, where the Weighted F1 score increases from $0.594$ to a peak of $0.636$ when scaling from $N = 8$ to $N = 32$. For FACED, performance peaks at $N = 16$, while for SEED-V, performance remains highly stable, suggesting that a smaller expert pool is sufficient for this task. As training more experts increases the total model parameters and pre-training cost, these results suggest a clear trade-off. A moderate number of total experts, such as $N = 32$, appears to offer a robust balance between performance gains and computational efficiency.

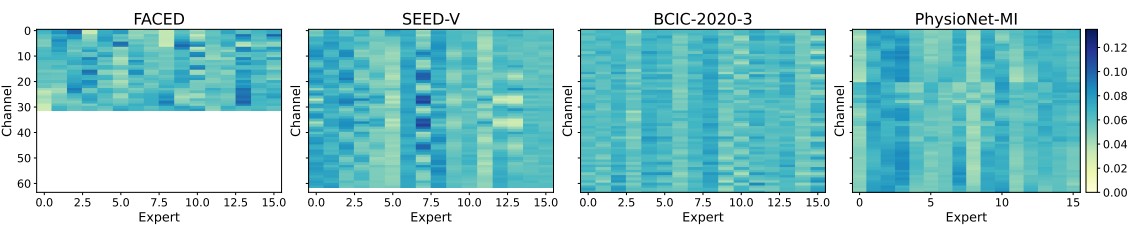

Figure 4: **Visualization of expert activation patterns** across different datasets. Each heatmap shows the average router probability for each channel (y-axis) across 16 experts (x-axis).

Figure 3b presents the performance with regard to the number of top-$K$ activated experts ($K \in \{1, 2, 4, 8\}$) from a fixed pool of 16 total experts. Overall, increasing $K$ improves or stabilizes performance, with the most notable gains observed on FACED and BCIC-2020. On FACED, the Weighted F1 score steadily increases from $0.625$ at $K = 1$ to $0.645$ at $K = 8$. For SEED-V and PhysioNet-MI, performance remains relatively stable across different values of $K$, suggesting that activating just one or two experts is sufficient to capture the necessary patterns for these tasks. These results highlight the flexibility of top-$K$ routing: while a modest $K$ is effective for some datasets, a larger $K$ can enhance expert collaboration and improve generalization on more complex tasks, allowing the model to dynamically adjust its inference complexity.

**Expert Activation acorss EEG Channels.** To validate that our channel-wise MoE architecture learns task-specific expert specializations, we visualize the expert activation patterns for different downstream tasks. Figure 4 displays heatmaps of the average router weights for each channel across all 16 experts. A key observation is that the activation patterns are distinctly different across the four datasets, providing strong evidence that BRAINMOE routes information to different experts based on the task. For instance, on the SEED-V dataset, we observe strong vertical bands, indicating that a small subset of experts (*e.g.*, experts 6 and 7) are consistently selected for specific groups of channels. In contrast, the activation for the BCIC-2020 imagined speech task is more distributed, suggesting that this complex task may require collaboration among a wider range of experts. These visualizations confirm that our model does not suffer from routing collapse. Instead, it successfully learns to dynamically allocate computational resources by leveraging a diverse set of specialized experts tailored to the unique neural patterns of different BCI tasks.

## 4    CONCLUSION

In this work, we introduced BRAINMOE, a novel foundation model that addresses the limitations of monolithic EEG architectures by employing a channel-wise MoE framework to adapt to the brain's inherent functional heterogeneity. Extensive experiments demonstrated that BRAINMOE consistently achieves new state-of-the-art performance across a diverse range of BCI benchmarks, outperforming strong foundation model baselines. The success of BRAINMOE establishes a promising new paradigm for building more efficient, and powerful EEG foundation models, opening new avenues for future research into interpreting expert functions and scaling towards the next generation of universal brain-computer interfaces.

**Limitations.** Despite its strong performance, BRAINMOE has several limitations that present opportunities for future work. Firstly, our pre-training is conducted on a corpus with relatively standardized channel configurations and segment lengths. This may limit its generalization to EEG modalities with drastically different spatial arrangements. Secondly, while effective, the MAE objective focuses primarily on signal reconstruction in the time domain, potentially overlooking richer features in the frequency or connectivity domains that are critical for many BCI tasks. Exploring complementary self-supervised objectives, such as contrastive learning, could yield more comprehensive representations.

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

## A RELATED WORK

**EEG Foundation Models.** Inspired by the success of foundation models in computer vision and natural language processing, a growing body of research is shifting beyond traditional task-specific EEG models (Lawhern et al., 2018; Song et al., 2022) towards developing generalized EEG foundation models. These efforts broadly fall into two categories based on their self-supervised pre-training objectives. One line of research leverages contrastive learning, exemplified by the pioneering BENDR model (Kostas et al., 2021) which learns representations from massive EEG data. Building on this, Brant (Zhang et al., 2023; Yuan et al., 2024) extends multi-signal learning for cross-modal representation. Another significant direction focuses on reconstruction-based objectives. BIOT (Yang et al., 2023) introduces a unified Transformer to tokenize biosignals into sentence-like structures, enabling cross-dataset pre-training. LaBraM (Jiang et al., 2024) segments EEG signals into channel patches and employs a neural tokenizer with vector-quantized spectrum prediction. More recently, CodeBrain (Ma et al., 2025) combines a dual-domain tokenizer for temporal and frequency components with an EEG-structured state-space model to capture complex dependencies, while EEGPT (Wang et al., 2024a) offers a pretrained Transformer for universal and reliable EEG representations. Our BRAINMOE contributes to this landscape by introducing a novel channel-wise Mixture-of-Experts architecture, specifically designed to address the functional heterogeneity of EEG channels by dynamically routing information to specialized experts.

**Mixture-of-Experts Models.** The Mixture-of-Experts (MoE) paradigm increases model capacity without proportionally raising computational costs by sparsely activating a subset of parameters, or "experts," for each input. First introduced in LSTMs by Shazeer et al. (2017), MoE layers are now commonly used to replace feed-forward networks in Transformers. The development of efficient routing strategies, such as Top-1 routing in Switch Transformer (Fedus et al., 2022) and Top-2 gating in GShard (Lepikhin et al., 2020), has made MoE a cornerstone of modern state-of-the-art large language models. However, these established MoE models were designed for NLP and perform routing on a token-by-token basis. This approach is ill-suited for EEG signals, as it would ignore the shared spatial and functional context of all temporal patches originating from a single channel. To address this, BRAINMOE introduces a novel **channel-wise** routing mechanism. By learning a holistic representation for each channel and routing all its constituent patches to the same experts, our approach respects the unique structure of brain signals, representing a key advancement for applying MoE to this domain.

## B TRAINING DETAILS

For pre-training data preparation, we adopt the same segmentation protocol as CBraMod (Wang et al., 2025) to ensure a fair comparison and leverage established data handling practices. The decoder layer to reconstruction the EEG signals is also the same as CBraMod. For the main experiments and comparison with SoTA methods, we configure BRAINMOE with a total of $N = 64$ experts, of which $K = 8$ are activated for each input. This configuration results in a model with approximately $150M$ total parameters, while only requiring the computation for $18M$ active parameters during inference, showcasing the efficiency of the MoE architecture. The number of learnable query tokens in the Channelformer is set to four, based on our ablation findings. To facilitate efficient experimentation in our ablation studies, we utilize a smaller configuration with $N = 16$ total experts and $K = 4$ activated experts, which reduces the inference computation to only $5M$ parameters. During the fine-tuning stage for downstream tasks, we adhere to the experimental settings established by CodeBrain (Ma et al., 2025) to ensure consistency and reproducibility. Across all experiments, the weight for the expert balance loss is consistently set to $\alpha = 0.01$ during both the pre-training and fine-tuning stages to prevent routing collapse and encourage expert specialization.

Table 3: **Performance comparison of different MoE configurations (Patch-wise, Temporal-wise, and Channel-wise Routing)** across datasets.

| MoE Type | Patch-wise | | | Temporal-wise | | | Channel-wise | | |
|---|---|---|---|---|---|---|---|---|---|
| Dataset | B-Acc | Kappa | F1-W | B-Acc | Kappa | F1-W | B-Acc | Kappa | F1-W |
| FACED | 60.45 | 55.27 | 60.46 | 63.71 | 58.80 | 63.66 | 62.94 | 58.10 | 62.98 |
| SEED-V | 41.27 | 27.33 | 42.37 | 41.30 | 26.97 | 41.96 | 41.41 | 27.81 | 42.27 |
| BCIC-2020 | 57.07 | 46.33 | 57.04 | 60.13 | 51.17 | 60.12 | 62.00 | 52.50 | 61.95 |
| PhysioNet-MI | 63.90 | 51.86 | 64.09 | 63.00 | 50.67 | 63.28 | 64.23 | 52.30 | 64.37 |
| **Average** | 55.67 | 45.20 | 55.99 | 57.04 | 46.90 | 57.25 | 57.64 | 47.68 | 57.68 |

## C    MORE EXPERIMENTAL RESULTS

**Analysis of MoE Routing Configuration.** We further investigate the optimal routing strategy for EEG signals by comparing three MoE configurations: 1) **Patch-wise routing**, where each EEG patch selects experts independently, akin to standard MoE in NLP; 2) **Temporal-wise routing**, where all channels at the same time step share the same experts; and 3) our proposed **Channel-wise routing**, where all patches within a single channel are routed to the same experts. This experiment aims to determine which structural prior is most beneficial for EEG decoding.

As shown in Table 3, the results demonstrate that enforcing structural consistency in routing is beneficial, with our proposed Channel-wise approach achieving the best overall performance. On average, Channel-wise routing outperforms both Patch-wise and Temporal-wise routing across all metrics. The superiority of Channel-wise routing is particularly evident on tasks like motor imagery (PhysioNet-MI) and imagined speech (BCIC-2020), where consistent activity within specific functional channels (*e.g.*, sensorimotor or language areas) is critical for decoding. While Temporal-wise routing shows some benefits over the unstructured patch-wise approach, it is less effective than grouping by channel. This experiment confirms that aligning the expert routing strategy with the functional organization of the brain, by channel, is a more effective design for EEG foundation models, validating our architectural choice for BRAINMOE.

**Robustness and Stability Analysis.** To assess the robustness of our proposed model, we evaluated the performance variance of the final Channel-wise MoE configuration across five independent runs with different random seeds. The standard deviations for the B-Acc metric across the four datasets (FACED, SEED-V, BCIC-2020, and PhysioNet-MI) were consistently low, at $0.72\%$, $0.76\%$, $0.83\%$, and $0.69\%$, respectively. Similarly, the standard deviations for Kappa and F1-W metrics remained under $1\%$ for all tested datasets.

These small values indicate that BRAINMOE produces highly stable and reproducible results, minimizing the impact of random factors like weight initialization or data shuffling. This stability can be attributed to the structured nature of our channel-wise MoE architecture. By consistently routing functionally related information from the same channel to a specific set of experts, the model learns stable and specialized representations, reducing the performance fluctuations that can affect less constrained models. This analysis demonstrates that BRAINMOE is not only highly accurate but also a reliable and robust model, a critical attribute for real-world Brain-Computer Interface applications where consistency is paramount.

**Effect of the Shared Expert.** To evaluate the contribution of the shared expert component, we conduct an ablation study comparing our final model against a variant where all experts are specialized and none are shared. The results, summarized in Table 4, demonstrate a clear and consistent benefit to including a shared expert. The model with a shared expert outperforms its counterpart on all four datasets across all metrics. On average, this leads to an improvement of approximately $1.5\%$ in both B-Acc and F1-W scores. The most

Table 4: **Ablation study on the effect of including a shared expert.**

| Dataset | No Shared Expert | | | Shared Expert (Ours) | | |
|---|---|---|---|---|---|---|
| | B-Acc | Kappa | F1-W | B-Acc | Kappa | F1-W |
| PhysioNet-MI | 64.1 | 52.2 | 64.3 | 64.6 | 52.7 | 64.9 |
| FACED | 60.7 | 55.6 | 60.8 | 63.6 | 58.9 | 63.8 |
| SEED-V | 40.7 | 26.3 | 41.5 | 41.9 | 27.9 | 42.8 |
| BCIC-2020 | 62.7 | 53.3 | 62.7 | 63.8 | 54.0 | 63.8 |
| **Average** | 57.0 | 46.8 | 57.3 | 58.5 | 48.4 | 58.8 |

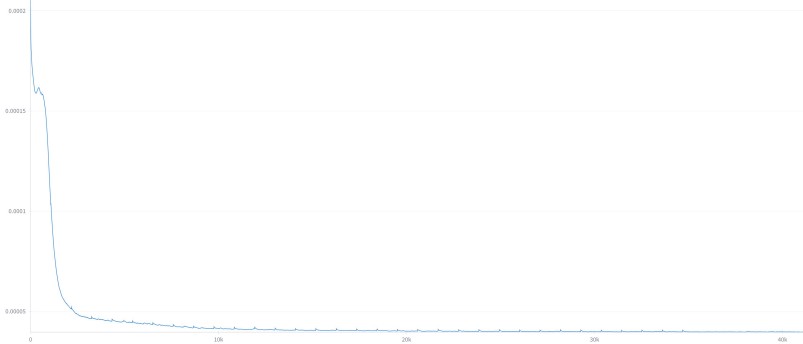

Figure 5: **Pre-training expert balance loss curve** for BRAINMOE.

notable gain is observed on the FACED dataset, where the B-Acc increases by nearly 3 percentage points. We hypothesize that the shared expert learns to model fundamental, ubiquitous patterns present across all EEG channels, such as common signal properties or baseline neural activity. By offloading this responsibility, the specialized experts are free to focus on learning the more nuanced, channel-specific features that are critical for discrimination. This effective division of labor results in a more robust and powerful representation, confirming that the inclusion of a shared expert is a valuable design choice for our BRAINMOE architecture.

**Importance of Expert Balance Loss.** A critical component in training MoE models is the auxiliary expert balance loss, designed to prevent routing collapse, where the gating network consistently favors a small subset of experts. To demonstrate its necessity, we conducted an ablation where we set the weight of this loss to zero. In this configuration, we observed a catastrophic failure of specialization: the router learned to select the exact same top-4 experts for nearly every channel, regardless of the downstream dataset. This effectively negates the benefit of the MoE architecture, reducing it to a smaller model with many unused parameters.

By introducing a small weight of $\alpha = 0.01$ for the auxiliary loss, we encourage the router to distribute the load more evenly across all available experts, forcing them to learn distinct and useful representations. The effectiveness of this combined objective is reflected in our pre-training loss curve, shown in Figure 5. The curve demonstrates stable and consistent convergence, with a rapid initial decrease as the model learns the primary reconstruction task, followed by a steady refinement of both reconstruction and expert balancing. The small periodic spikes in the later stages of training are characteristic of learning rate scheduler resets, but the overall downward trend confirms successful optimization. Therefore, the expert balance loss is



Figure 6: **Visualization of expert activations across layers** for four EEG datasets.

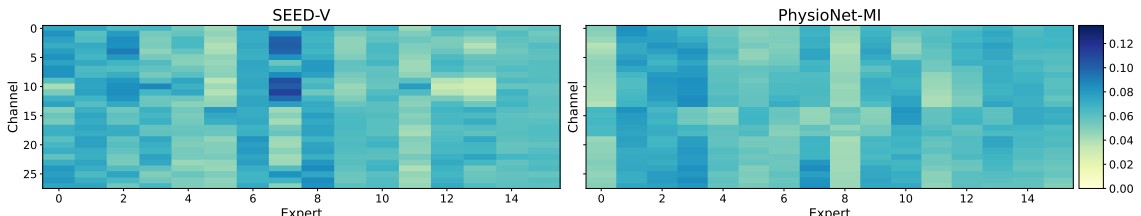

Figure 7: **Comparison of expert activation** between SEED-V and Physio-Net datasets.

an essential component for enabling true specialization and realizing the full potential of the BRAINMOE architecture.

**Expert Activation Across Layers.** To further understand how experts are utilized throughout the network, we visualize the average expert activation probabilities for each layer. Figure 6 displays heatmaps where the y-axis represents the network layer (from 0 to 11) and the x-axis represents the 16 experts. The color intensity indicates the average activation weight, aggregated across all channels for a given task.

A striking observation is the remarkable consistency of expert activation patterns across all layers within each specific task. This is evidenced by the strong vertical bands in the heatmaps, which show that if an expert is highly activated in the initial layers, it tends to remain highly activated in the deeper layers as well. While the set of preferred experts clearly differs from one dataset to another (*e.g.*, the prominent experts for SEED-V are different from those for PhysioNet-MI), this task-specific "expert signature" remains stable throughout the network's depth. This suggests that BRAINMOE does not learn layer-specific specializations (*e.g.*, some experts for low-level features and others for high-level features). Instead, it appears to learn a holistic, task-dependent routing strategy that is applied consistently at every stage of processing, reinforcing the idea of a stable and specialized computational pipeline for each unique BCI task.

**Task-Specific Expert Routing on Shared Channels.** To further investigate the task-specific nature of the learned specializations, we conduct a direct comparison of expert activation patterns between the SEED-V (emotion recognition) and PhysioNet-MI (motor imagery) datasets. This analysis is particularly insightful as the first 29 channels in both datasets correspond to the same physical scalp locations, allowing us to isolate the effect of the cognitive task on the expert routing mechanism.

As illustrated in Figure 7, the expert activation profiles are strikingly different between the two tasks, even for the same channels. For the SEED-V dataset, a specific subset of experts—most prominently experts 7 and 8—are strongly and consistently activated across numerous channels, suggesting they have become specialized in processing signals pertinent to emotional states. In stark contrast, the heatmap for PhysioNet-MI reveals a different and more distributed activation profile. Here, experts such as 1, 3, 9, and 10 are frequently selected. Notably, while expert 7 is also active in this task, it is part of a broader coalition of

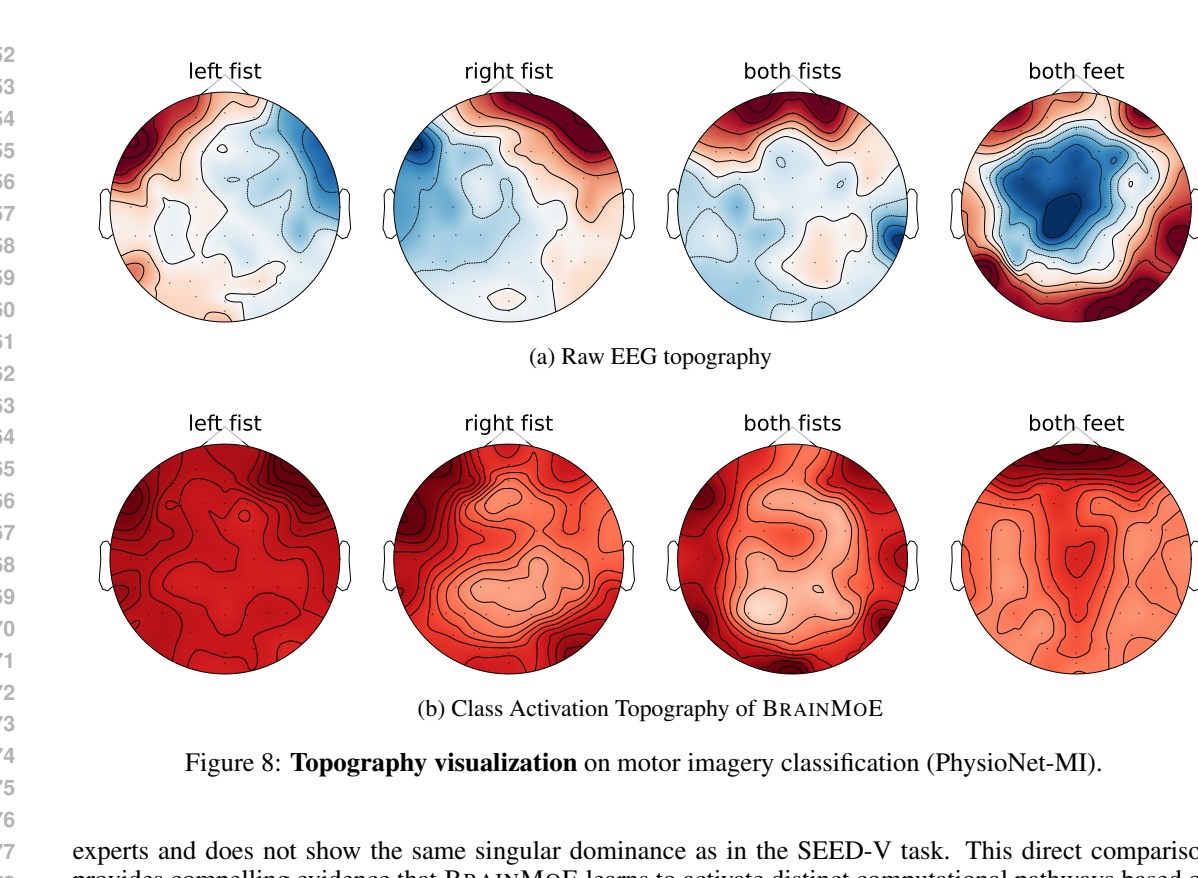

(a) Raw EEG topography

(b) Class Activation Topography of BRAINMOE

Figure 8: **Topography visualization** on motor imagery classification (PhysioNet-MI).

experts and does not show the same singular dominance as in the SEED-V task. This direct comparison provides compelling evidence that BRAINMOE learns to activate distinct computational pathways based on the task at hand, rather than just learning static, channel-specific features.

**Topography Visualization.** To investigate whether BRAINMOE learns neurophysiologically plausible spatial features, we visualize its class activation topographies using Grad-CAM on the PhysioNet-MI dataset. Figure 8 compares the raw EEG topography for four motor imagery classes (left fist, right fist, both fists, and both feet) with the corresponding class activation maps generated by our model.

The activation maps in Figure 8b demonstrate a remarkable alignment with established neurophysiological principles. For the imagery of a "left fist" movement, BRAINMOE shows the highest activation over the right hemisphere, corresponding to the contralateral motor cortex. Conversely, for the "right fist", the activation is correctly localized over the left hemisphere. The imagery of "both feet" results in a strong, focused activation at the vertex (top-center of the scalp), which is consistent with the location of the leg and foot representation in the motor homunculus. These patterns learned by the model are not arbitrary; they closely mirror the underlying brain activity visible in the raw EEG signals shown in Figure 8a. This strong correspondence confirms that BRAINMOE is not merely fitting to statistical artifacts but is learning to identify and leverage class-discriminative information from the correct, functionally relevant brain regions, thereby validating the quality and interpretability of its learned representations.

## D   THE USE OF LARGE LANGUAGE MODELS (LLMS)

In the preparation of this manuscript, a large language model (LLM) was utilized as a writing assistant. The model's role was strictly limited to improving the quality of the written text. Specifically, it was employed

for tasks such as correcting grammatical and spelling errors, refining sentence structure for better clarity, and polishing the language to enhance overall readability. The LLM was not used for any core scientific or conceptual aspects of this research. All elements of the research, including the ideation of the BRAINMOE architecture, the literature review, the experimental design, the analysis of results, and the interpretation of findings, were conducted entirely by the human authors. The authors bear full responsibility for all scientific contributions and the final content of this paper.

