# OpenReview forum: "BrainMoE: Towards Universal EEG Foundation Models with Channel-wise Mixture-of-Experts"
_ICLR.cc/2026/Conference — ICLR 2026 Conference Withdrawn Submission_

### Official Review · Reviewer_J6Yf · 2025-10-20

**Soundness:** 3
**Presentation:** 4
**Contribution:** 3
**Rating:** 4
**Confidence:** 4

**Summary:**

This paper introduces the famous MOE technique for a transformer-based pretraining for EEG. Downstream task performances as well as several ablations and analyses support the gains from this. The paper also introduces ChannelFormer for properly encode each channel to be routed for k experts.

**Strengths:**

- Paper is clear to follow with great figure and table presentations.
- Incorporation of MoE for channel-wise routing seems domain specific and well adapted.
- Comparison and Ablation studies are thorough, including several state of the art.
- Correctly identifies limitations in the conclusion section

**Weaknesses:**

- Motivation not so well supported: lines 54-57 are not well supported, as to what it means by "specialized functions of different brain regions". This claim requires an interpretation study in Figures 4,6,7 to be improved, including more application driven analyses, in relation to disease or known pathways.
- Related to the above point, the relative gain from MoE integration seems marginal.
- Novelty and technical contributions are marginal.

**Questions:**

- Would MoE integration to other baselines surpass this model, judging by that the ablation without MoE performs much worse?
- Also related to above question, integrating MoE seems possible for most modern architectures with minimal effort.
- Will this method generalize to electrode locations unseen during training?
- Can you show the expert coupled with the spatial location of those EEGs? Do experts specialize in space or time? Do the expert distribution also vary in task e.g., resting state vs. motor imagery. These interpretations may further strengthen the novelty claim of this paper.

---

### Official Review · Reviewer_nBSh · 2025-10-22

**Soundness:** 3
**Presentation:** 3
**Contribution:** 2
**Rating:** 4
**Confidence:** 4

**Summary:**

The paper introduces BrainMoE, a universal EEG foundation model that integrates a channel-wise Mixture-of-Experts (MoE) architecture. The key idea is to dynamically assign specialized experts to individual EEG channels, reflecting the brain’s functional heterogeneity. The model employs a ChannelFormer to learn channel-level embeddings, followed by a channel-wise MoE block that activates the most relevant experts selected by a router to learn spatial information. Pre-trained using a masked EEG reconstruction (MAE) objective on the Temple University Hospital EEG corpus, BrainMoE achieves new state-of-the-art performance on several EEG decoding benchmarks (emotion recognition, motor imagery, sleep staging, seizure detection, etc.). The paper also includes detailed ablations and visualizations showing expert specialization patterns.

**Strengths:**

1. The introduction of a channel-wise MoE represents a creative adaptation of sparse expert models to EEG processing, capturing spatial interactions.

2. The manuscript is well organized and features high-quality figures.

3. The experiments span nine datasets and seven EEG decoding tasks, demonstrating BrainMoE’s strong generalization ability.

4. Visual analyses of expert activations across channels and layers provide insights into task-dependent routing, a step toward interpretability in large EEG models.

**Weaknesses:**

1. **[Major]** Although introducing MoE for spatial learning is interesting, it is the only major novelty of this paper. The other functional blocks in the framework are either existing components or incremental adaptations.

2. **[Major]** There is no novel pre-training strategy; the model uses the commonly adopted mask-and-reconstruction (MAE) objective, which limits methodological innovation.

3. **[Major]** No standard deviation (std) is reported in the experimental evaluation. Previously published EEG foundation models typically report results averaged over five random seeds. The lack of multiple runs prevents proper assessment of generalization and robustness.
   - Have you run the experiments using different seeds?
   - If so, how were these seeds selected?
   - Random seed choice can significantly affect performance [a].
   [a] Liu, Huan, et al. *“LibEER: A comprehensive benchmark and algorithm library for EEG-based emotion recognition.”* IEEE Transactions on Affective Computing (2025).

4. The evaluation metrics differ between Table 1 and the analysis experiments.

5. For the study of MoE configuration (Figure 3), **balanced accuracy (balanced ACC)** is missing and would be more appropriate for datasets with balanced classes.

6. There is no discussion of model size or computational complexity (FLOPs) compared to baselines.

7. No reproducible code or pretrained weights are provided.

8. Hyperparameter tables for pre-training and fine-tuning are missing.

**Questions:**

1. Can the authors clarify what specific new design elements, beyond the channel-wise MoE, distinguish BrainMoE from previous foundation models such as LaBraM, BIOT, or CBraMod?

2. What motivates the architectural choices for the other functional modules — were they directly reused or substantially modified?

3. Were all experiments conducted with a single random seed? If not, how were the seeds chosen?

4. Could the authors report the mean ± standard deviation of the performance metrics over multiple runs?

5. If experiments were conducted only once, how can readers trust the robustness of the reported results?

6. Why are the evaluation metrics (e.g., accuracy, F1-score, balanced accuracy) inconsistent across tables?

7. Why was balanced accuracy (balanced ACC) not included in Figure 3 for the MoE configuration study?

8. What are the total parameters, active parameters during inference, and FLOPs of BrainMoE compared to other EEG foundation models?

9. Do the authors plan to release the code and pretrained weights for reproducibility? Without such resources, how can the community validate or extend BrainMoE?

10. Could the authors include detailed hyperparameter settings (e.g., learning rate, masking ratio, optimizer type, batch size, dropout rate)?

**Details Of Ethics Concerns:**

N.A.

---

### Official Review · Reviewer_XwSu · 2025-10-28

**Soundness:** 2
**Presentation:** 3
**Contribution:** 2
**Rating:** 2
**Confidence:** 4

**Summary:**

The paper introduces a new foundation model for EEG signals. Its innovation lies in the design of dynamic assignment of channel-wise mixture-of-experts to different channels for better decoding.

**Strengths:**

The paper is based on the concept that better EEG foundation models will come from designing mechanisms that can be tailored to the various pre-training tasks instead of having a uniform data structure for all tasks. The authors claim that a mixture-of-channel-wise-experts is the potential answer to this identified problem / hypothesis and they provide a series of experiments, comparisons and ablation studies to make that point.

**Weaknesses:**

The paper starts with some strong statements: claiming that segmentation of EEG into patches and “monolithic” transformer layers are used in the current foundation models but yet these are exactly the same blocks / tools the paper uses in the proposed method. In my view, the whole methodology keeps many elements from CBraMod and adds some complex building blocks without significant innovation or theoretical benefits. From the paper, I am not convinced how only channel information is so significant for a foundation model trained on only TUEG. The paper does not provide any signal reconstruction metrics or samples. Important building blocks are not even described in the main paper (like the decoder for which the reader needs to go to the appendix). The paper also misses important implementation details: e.g. how the fine-tuning of the models has been performed. The activations of Figure 4 do not provide much interpretable insights - but a strong signal that the model overfits on different datasets.

Writing:
The paper also needs some re-writing. Section 2.1 is a currently very complicated to understand without reading it a few times. For example, h is introduced in 130 line and the reader needs to reach line 139 to understand the connection with e. Also some grammar / English checks in the paper are needed - for example “We replaces” line 126. Related work could be moved in the main paper rather than the Appendix.

Overall:
It feels that this work is still at an early stage. Feels an over-complicated method with lack of evidence of its benefit. The paper needs to be updated and more evidence needs to be added to convince of its merit.

**Questions:**

1. How has the fine-tuning process taken place for all models (baselines and foundation models) ? The difference is so small that we cannot be sure that this is an actual benefit or just unfair fine-tuning for some models
2. Which layers are fine-tuned ? In Figure 4, have the experts / router been fine-tuned ?
3. If the whole foundation model has been fine-tuned along with the classification layer, have the authors tried experiment where the foundation models are frozen and the classification layer only is getting trained?
4. How about the reconstruction ? Any examples here.

---

### Official Review · Reviewer_CUHN · 2025-10-30

**Soundness:** 1
**Presentation:** 3
**Contribution:** 1
**Rating:** 2
**Confidence:** 4

**Summary:**

The paper introduces BrainMoE, a novel channel-wise Mixture-of-Experts (MoE) EEG foundation model. This architecture is designed to dynamically assign specialized processing experts to different EEG channels. The authors posit that this approach enables task-adaptive decoding, claiming superior performance over prior models across multiple diverse downstream benchmarks.

**Strengths:**

- The paper features clear and well-designed figures that effectively illustrate the proposed BrainMoE architecture and its core concepts.

- The authors conduct comprehensive experiments across nine diverse downstream tasks, validating the empirical effectiveness and initial generalizability of BrainMoE as an EEG foundation model.

**Weaknesses:**

- There is a significant lack of logical connection and justification between the stated problem (distinct neural circuit activation across tasks) and the proposed solution (channel-wise MoE expert assignment). Specifically, the model uses channel embeddings $\mathbf{c}_i$, derived solely from the entire temporal sequence of each channel, to determine the informational content and importance of a brain region for a specific task. It is not sufficiently justified, either theoretically or empirically, that the univariate temporal dynamics of a single channel provide the necessary and sufficient information to determine its critical role in a task for expert routing. This methodological choice fundamentally undermines the persuasiveness of the expert assignment mechanism.

- The absence of statistical significance tests (e.g., $p$-values or confidence intervals) fundamentally weakens the evidence supporting the claimed effectiveness of the proposed method over baselines. Observed performance gains may be due to chance.

- The inconsistent and poorly justified use of evaluation metrics across the paper compromises the reliability and fairness of the reported results. Specifically, Table 1 uses Balanced Accuracy and Weighted F1; Figure 2 and Table 2 add Cohen’s Kappa; and Figure 3 uses only Cohen’s Kappa and Weighted F1. Furthermore, the Evaluation Metrics section on page 5 entirely omits any mention of Cohen’s Kappa.

- The comparison against referenced models (LaBraM and CBraMod) is potentially unfair. Foundation model performance typically scales with size, a trend noted in EEG models [1]. BrainMoE contains 150 M total parameters (with 18 M active), whereas the baselines reportedly have only 4–5 M parameters. Performance evaluation at a comparable parameter scale is essential to clarify whether the performance gain is attributable to the novel MoE architecture itself or simply the increased model size.

**Questions:**

- On page 4, the dimension of the input feature $\mathbf{W}_j$ is defined. What is the explicit shape of the weight matrix $W_j$ used in the routing mechanism?

- Was the proposed method exclusively evaluated in a subject-independent setting (i.e., cross-subject validation)? If so, how does the model perform in a subject-dependent setting (i.e., within-subject validation)? Furthermore, is the MoE architecture's efficacy maintained or diminished under these subject-dependent scenarios?

---

### Note · Authors · 2026-01-17

I have read and agree with the venue's withdrawal policy on behalf of myself and my co-authors.